# Explorative Meta-Analysis of 417 Extant Archaeal Genomes to Predict Their Contribution to the Total Microbiome Functionality

**DOI:** 10.3390/microorganisms9020381

**Published:** 2021-02-13

**Authors:** Robert Starke, Maysa Lima Parente Fernandes, Daniel Kumazawa Morais, Iñaki Odriozola, Petr Baldrian, Nico Jehmlich

**Affiliations:** 1Laboratory of Environmental Microbiology, Institute of Microbiology of the Czech Academy of Sciences, 142 00 Praha, Czech Republic; maysaparente1@gmail.com (M.L.P.F.); daniel.morais@brmicrobiome.org (D.K.M.); inaki.odriozola@biomed.cas.cz (I.O.); baldrian@biomed.cas.cz (P.B.); 2Molecular Systems Biology, Helmholtz-Center for Environmental Research, UFZ, 04318 Leipzig, Germany; nico.jehmlich@ufz.de

**Keywords:** archaea, functional diversity, microbiome functionality

## Abstract

Revealing the relationship between taxonomy and function in microbiomes is critical to discover their contribution to ecosystem functioning. However, while the relationship between taxonomic and functional diversity in bacteria and fungi is known, this is not the case for archaea. Here, we used a meta-analysis of 417 completely annotated extant and taxonomically unique archaeal genomes to predict the extent of microbiome functionality on Earth contained within archaeal genomes using accumulation curves of all known level 3 functions of KEGG Orthology. We found that intergenome redundancy as functions present in multiple genomes was inversely related to intragenome redundancy as multiple copies of a gene in one genome, implying the tradeoff between additional copies of functionally important genes or a higher number of different genes. A logarithmic model described the relationship between functional diversity and species richness better than both the unsaturated and the saturated model, which suggests a limited total number of archaeal functions in contrast to the sheer unlimited potential of bacteria and fungi. Using the global archaeal species richness estimate of 13,159, the logarithmic model predicted 4164.1 ± 2.9 KEGG level 3 functions. The non-parametric bootstrap estimate yielded a lower bound of 2994 ± 57 KEGG level 3 functions. Our approach not only highlighted similarities in functional redundancy but also the difference in functional potential of archaea compared to other domains of life.

## 1. Introduction

The biochemical transformations conducted by a community of microbes from all domains of life mediate ecosystem functioning [1]. Even though ecological studies tend to focus on bacteria and fungi, archaea as a major part of global ecosystems [2] are ubiquitous in both terrestrial and aquatic environments [3,4]. Particularly, archaea make up between 20% and 30% of the total prokaryotes in marine environments [5] and between 0% and 10% in soil environments [4,6]. Increases in the proportion of archaea were found in extreme habitats such as acidity and low temperature [7]. Functionally, archaea play key roles in global carbon (e.g., methanogenesis or CO_2_ fixation) and nitrogen (e.g., N_2_ fixation or oxidation of ammonia) cycles [8], but they also have complex relationships with both bacteria and fungi [9]. In a microbial community, multiple organisms with different taxonomy may have similar if not identical roles in ecosystem functionality, the so-called functional redundancy [10]. Indeed, interspecies redundancy was reported to be very high with several hundreds to thousands of different taxa to express the same function in a habitat [11]. These functions can be statistically inferred based on the homology to experimentally characterized genes and proteins in specific organisms to find orthologs present in a microbiome [12,13,14]. This ortholog annotation is used by KEGG Orthology [15,16], which covers a wide range of functional classes (level 1 of KEGG) comprising cellular processes, environmental information processing, genetic information processing, human diseases, metabolism, organismal system, brite hierarchies, and functions not included in the annotation of the two databases pathway or brite. KEGG level 2 functions provide more detail, i.e., differentiating glycoside hydrolases and glycosyltransferases within carbohydrate active enzymes (level 1), whereas KEGG level 3 is the enzyme itself, i.e., the glycogen phosphorylase (K00688, EC: 2.4.1.1.) that belongs to the glycosyltransferases (more information can be obtained under https://www.genome.jp/kegg/kegg3.html, accessed on 17 July 2020). However, the bottleneck of reporting microbiome functions is the low number of fully sequenced and annotated genomes as only organisms are captured that have undergone isolation and extensive characterization [12,13,14] with respect to the expected total diversity. Hence, the lower the share of known species or the higher the predicted total diversity, the weaker the prediction itself. Problematically, the vast majority of organisms have not yet been studied [17,18] which is why the annotation is based on the similarity to the genomes of the very few studied model organisms [12,13,14]. Consequentially, microbiome functionality is inferred based on the taxonomic composition of the community and its relation to functional parameters [19], indicated by the frequent use of the 16S rRNA gene metabarcoding to describe the prokaryotic community. Even though the description of microbial communities is important to assess the drivers of the occurrence of individual taxa and the composition of their communities [12], the mere taxonomic composition itself did not provide detailed answers about its functional diversity [20]. The functional diversity for both bacteria [13] and fungi [12,13,14] were recently predicted to comprise millions of different functions using meta-analyses of proteins [13] and genomes [12,14], most of which are unknown today. However, our understanding of functional redundancy in archaea and their contribution to the total microbiome functionality is still scarce. 

Here, we used both parametric and non-parametric estimators of functional richness with the aim to predict the total archaeal functionality on Earth and to unveil the relationship between taxonomy and function in the archaeal domain. To do so, we obtained all completely annotated genomes of taxonomically unique archaeal species (n = 417) from the integrated microbial genomes and microbiomes (IMG) of the Joint Genome Institute (JGI) (https://img.jgi.doe.gov/, accessed on 17 July 2020) with taxonomic annotation on the species level and functional annotation of KEGG on level 3 (referred to as KEGG function). We used a parametric estimation based on accumulation curves (AC) [21] that are characterized by the increasing number functions with increasing species. The AC was fitted to saturated, unsaturated, and logarithmic models, and the best fitting model was chosen based on its fitness in comparison to the other models. As a non-parametric estimator, Chao-1 was used for every 50 randomly picked species of all 417 in the database each with 20 replicates. The precision of both the parametric and the non-parametric approach generally depends on the proximity to the asymptote of the model, with greater extrapolation to the total count resulting in greater error [22]. We therefore hypothesized more precise estimates of the contribution of archaea to the total microbiome functionality than previously proposed for both bacteria [13] and fungi [12,13,14] due to the higher coverage of the predicted taxonomic diversity in archaea.

## 2. Materials and Methods

### 2.1. Metadata Collection of the Total Known Archaeal Microbiome Functions

To predict the contribution to microbiome functions and to compare the genome content across archaeal species, habitats, and temperature ranges, available genomes from archaea were downloaded from the integrated microbial genomes and microbiomes (IMG) of the Joint Genome Institute (JGI) on 17 July 2020. A genome was randomly selected in the case of multiple sequenced genomes from the same species to obtain taxonomically distinct archaeal species. For each genome, the gene counts for each KEGG function [15,16] were retrieved. Our database comprised 417 completely annotated archaeal genomes with, in total, 2835 KEGG functions (Appendix A). Noteworthy, 761 archaeal metagenome-assembled genomes (MAG) were available in the database (as of 31 August 2020) but only seven with high quality. Even though many archaeal genomes and functions are derived from non-cultivable species, we wanted to use complete information for precise modeling. For one genome, the sequencing status was denoted as “Draft”, for 217 as “Permanent Draft”, and for 199 as “Finished”. Only three genomes were available to describe psychrophilic archaea and the taxa were therefore excluded from further analysis. Intergenome redundancy was calculated as the number of KEGG functions covered by one randomly chosen species in comparison to the total number of functions in all species [12]. Intragenome redundancy or gene redundancy was estimated as the average of genes per individual KEGG function in one species [12]. The gene counts and KEGG functions per archaeal phylum, habitat, and temperature range were retrieved as the average with standard deviation from the database. To estimate the specific differences, both intergenome and intragenome redundancy were calculated for every phylum, habitat, and temperature range as described for the total database above.

### 2.2. Accumulation Curves (AC)

Archaeal species were added in intervals of one to 417 species using 1000 random permutations per step via the function *specaccum* from the R package *vegan* [23]. A saturated (Equation (1)) and an unsaturated model (Equation (2)) with the critical point estimated by the term 3*A_f_* [24] was then fitted to the AC of the database permutation. Due to the plateauing shape of the AC, a logarithmic model was used as well. The fit of all models was validated by the analysis of the Akaike Information Criterion (AIC) [25] with a penalty per parameter set to *k* equals two. The total number of KEGG functions in archaea on Earth was predicted using a global species richness estimate of 13,159 archaeal species [26] to calculate the potential maximum of KEGG functions via uncertainty propagation and Monte Carlo simulation of the function *predictNLS* in the R package *propagate* [27]. The non-parametric estimation of functional richness was calculated by Chao-1 [28,29]. This method was developed to estimate the asymptotic species richness in a set of samples. Since our objective was to estimate the asymptotic functional richness, genomes took the role of samples and KEGG functions took the role of species in our analysis. Resampling and repeating computations for lower levels of sample accumulation generated a smoother curve of the estimations. A reliable estimator would reach its own asymptote before the species accumulation curve does [21]. To test whether this occurred in our dataset, Chao-1 was estimated using a random subset of every 50 picked archaeal genomes in the database starting with two species (Equation (3)). Additionally, asymptotic functional richness was estimated using a first order jackknife (jack-1) and the bootstrap “boot” methods with the function *specpool* in the R package *vegan* [23] to check if these two alternative methods yielded estimations comparable to the parametric and Chao-1 estimates.
Functional richness = *f_max_* × Species richness/(*A_f_* + Species Richness),(1)
Functional richness = *f_max_* × Species richness/(*A_f_* + Species Richness) + (*k* × Species richness),(2)
Chao-1 = Functional richness + (a_1_^2^/2a_2_),(3)

Here, *f_max_* is the maximum functional richness, *A_f_* the accretion rate of functions with an increasing number of species, and *k* the constant of the additive term. Functions found only once or twice are indicated by a_1_ as singletons and a_2_ as doubletons, respectively.

### 2.3. Statistical Analysis

The differences between gene counts, KEGG functions, and their functional redundancy were estimated by Tukey’s honestly significant difference (HSD) test [30] using the package *agricolae* [31].

## 3. Results

### 3.1. Gene Counts and Number of KEGG Level 3 Functions

The gene count per species was significantly higher (HSD-test) in *Euryarchaeota* as compared to *Crenarchaeota* and *Thaumarchaeota* (Figure 1a). On the level of habitats, archaea isolated from fresh water, sediments, or soils had on average significantly more genes than archaea enriched from the deep sea or hot springs. A comparable number of archaeal genomes were sampled from each habitat, ranging from 8 in sludge to 37 in hot springs. On the level of temperature preferences, mesophilic archaea comprised significantly (HSD-test) more genes than thermophilic and hyperthermophilic archaea. Similar significant differences were apparent in the number of KEGG level 3 functions on all three prior investigated levels (Figure 1b).

### 3.2. Inter- and Intragenome Functional Redundancy

Intergenome functional redundancy is a proxy for the performance of one metabolic function by multiple taxonomically distinct organisms, while intragenome functional redundancy describes the number of replicated functions within one genome [12,14]. Across all 417 archaeal genomes, the median of intergenome functional redundancy was found to be 0.06 (Figure 2a). Most functions were found with low redundancy as 1650 KEGG functions were present in less than 10% of the species. In comparison, only 172 KEGG functions were present in more than 90% of the archaeal genomes. Together, 65.3% of all functions showed either high or low redundancy while the rest appeared intermediate with an intergenome functional redundancy between 0.1 and 0.9 with a particularly high abundance at around 0.24. The median of intragenome functional redundancy across all 417 archaeal genomes was found to be 1.02 gene copies per KEGG function with a maximum of 72 gene copies (Figure 2b). 

Among archaeal phyla, *Thaumarchaeota* showed a significantly higher (HSD-test) intergenome functional redundancy compared to *Crenarchaeota* and *Euryarchaeota* (Figure 3a). Within habitats, the intergenome redundancy in the deep sea, hot springs, sediments, and sludges was significantly higher (HSD-test) than in fresh water, host, marine, and soil habitats. On the level of temperature preferences, intergenome redundancy was highest in hyperthermophilic archaea, followed by thermophilic and mesophilic ones. The inverse pattern was found for intragenome functional redundancy for all three investigated levels (Figure 3b). Significantly higher intergenome redundancy was accompanied by significantly lower intragenome redundancy and vice versa regardless the taxonomy, habitat, and temperature preference of archaea.

### 3.3. Parametric and Non-Parametric Estimation of the Archaeal Contribution to the Total Microbiome Functionality

The logarithmic model comprised a significantly better fit of the dependence of functions on archaeal species richness than both the saturated and the unsaturated model, estimated by lower akaike information criterion (AIC) (Figure 4a) to imply a plateau of functional richness with higher species richness. Considering the estimate of 13,159 archaeal species on Earth [26] and assuming that the relationship between species richness and functional richness will be logarithmic with the addition of new species, we propagated the logarithmic model with the result of a total archaeal functionality of 4164.2 ± 2.9 KEGG functions (with 4158.6 and 4169.9 as 95% confidence intervals). Similarly, the non-parametric estimator of functional richness that assumes the existence of a maximum functional richness indeed plateaued for the 417 archaeal genomes (Figure 4b). Estimations obtained with more than 200 archaeal genomes generated broadly overlapping confidence intervals indicative of the reliability of the estimation of the asymptotic functional richness. The three non-parametric estimators yielded comparable estimations of asymptotic functional richness: 3128 ± 42 KEGG functions using the Chao-1 index, 3169 ± 78 using the first order jackknife, and 2994 ± 57 using the bootstrap method (Figure 4b).

## 4. Discussion

### 4.1. Genome Content

The genome size of an organism generally reflects its developmental and ecological needs [32]. Larger genomes are directly related to increases in both cellular and nuclear volumes [33] that help to cushion fluctuations in concentrations of regulatory proteins or to protect coding DNA from spontaneous mutation [34]. Variation in genome size is therefore a result of the adaptive needs or of natural selection in different organisms [32]; the so-called adaptive theory of genome evolution. The smaller genomes of archaea could be directly related to a higher evolutionary rate [35]. Indeed, the 417 archaeal species generally comprised smaller genomes compared to bacteria [36], but with statistically significant differences among phyla, habitats, and temperature ranges that were mirrored by the number of KEGG level 3 functions in each genome. Particularly, archaea inhabiting extreme habitats such as deep sea or hot springs characteristic with high local temperatures not only had significantly fewer total genes but also fewer KEGG functions. Otherwise, environments of higher complexity and diversity such as soils or sediments contained archaea with a larger functional potential that may have allowed them more options for the competition for or the utilization of a wider range of nutrients.

### 4.2. Functional Redundancy

A limited set of metabolic pathways found in a variety of taxonomic groups drive most biogeochemical reactions [37] which is why the diversity in the community is correlated strongly with its functional diversity [38]. Functions are classified into two groups [12,13,14]: (i) Highly redundant across different species present in more than 90% of all species or (ii) unique to only a few species present in less than 10% of all species. Here, intergenome redundancy was either high or low for roughly two thirds of all the KEGG functions; fewer than the 77.3% were found in fungal genomes [12,14]. However, the presence of a higher share of functions of intermediate redundancy that are present between the two thresholds suggested the presence of more than two groups [12,13,14] that could be particularly important for organisms with smaller genomes such as archaea and bacteria. A set of functions present in a quarter of all archaea indicated that the presence of a driving phylotype or environment may drive intergenome redundancy. Indeed, most functions (151/194) with an intergenome redundancy between 22% and 26% belonged to the phylum *Crenarchaeota*, the habitat hot springs, and the temperature range of hyperthermophilic archaea, mainly affiliated with amino acid utilization, fermentation, methanogenesis, and nucleic acid metabolism. The median intergenome redundancy was twice as high as found for fungi before [12,14], implying a higher share of functions shared among archaea on average. However, only half the gene copies (1.02 in archaea compared to 2.0 in fungi) were present, highlighting the close relationship between intergenome and intragenome redundancy. Indeed, the archaeal genomes revealed that low intergenome redundancy is generally related to high intragenome redundancy and vice versa. Presumably, every organism must choose between additional copies of functionally important genes or a higher number of different genes, especially in reduced genomes. Similarly to the pattern found in fungi before [12,14], functions belonging to the maintenance apparatus such as S-adenosylmethionine synthetase (EC 2.5.1.6, K00789) involved in the biosynthesis of amino acids were with both high intergenome and high intragenome redundancy, allowing for more complex regulation of the gene, i.e., when more transcripts are needed. Otherwise, functions with low intergenome and low intragenome redundancy are highly specialized processes only performed by a few archaea such as the drug transporter MFS transporter, DHA1 family, multidrug resistance protein (K19578) found in the crenarchaeote *Thermofilum adornatus*. 

### 4.3. Archaeal Contribution to the Total Microbiome Functionality

The parametric approach estimated the archaeal contribution to the total microbiome functionality to roughly 4200 KEGG functions; a magnitude less than predicted for both bacteria [13] and fungi [12,13,14]. The lower bound estimate of functional richness derived from the non-parametric approaches yielded roughly 3000 KEGG functions. A plateau of functional richness with higher species richness made the predictions for archaea more reliable as the errors decreased with proximity to the asymptote [22]. Theoretically, a higher number of species must be sequenced until no additional functions are unveiled and the accumulation curve reaches the actual asymptote [39]. However, practically, this is nearly impossible as a prohibitively large number of species are needed to be sampled in order to reach an asymptote [40]. In our meta-analysis, admittedly, the 417 genomes of distinct archaeal species only spanned three archaeal phyla from all 21 proposed phyla [41,42] and covered only a small part of the predicted taxonomic diversity in archaea; with databases containing up to 13,159 archaeal species [26], the prediction of 5000 archaeal genera [43], and the finding of 669 distinct archaeal species among 10,575 prokaryotic genomes [44]. Hence, the addition of genomes from novel archaeal species with potentially new KEGG functions could change both the parametric and the non-parametric estimates of functional richness. However, the differences in the estimates are likely not as tremendous as the potential differences in the estimates for both bacteria [13] and fungi [12,13,14] as the accumulation curve already plateaued with 417 taxonomically distinct archaeal species. Noteworthy, it is unclear how well new functions are recovered in archaea. As there is notably less interest in archaea compared to bacteria, functional annotations might generally miss archaea-specific functions to a larger extent than bacteria-specific functions missed in bacteria. As of today, our understanding of the contribution of archaea to the total microbiome functionality covers the majority of the KEGG functions, but many as-yet unknown archaea-specific functions could exist.

## 5. Conclusions

Our results suggest a limited contribution of archaea to the total functional potential of the microbiome, with most archaeal functions already identified as of today. However, the existence of archaea-specific functions must be validated by novel and more sophisticated methods. The accumulation curve describing the increase of functional categories with the number of sequenced genomes in archaea was closer to the asymptote than in bacteria [13] and fungi [12,13,14]. This made the estimate of archaeal contribution to the total microbiome functionality more precise, although it is still uncertain if the functional diversities of different domains can easily be compared. Noteworthy and similar to fungi, only one quarter of all genes in archaeal genomes on average were affiliated with a KEGG function, which demonstrates the limitations of the annotation because the prediction of microbiome functionality technically excluded three quarters of the entire functional potential in archaea. Different ortholog databases such as COG or Pfam could further improve our understanding of functional diversity, especially in archaea, as those covered three times more genes than KEGG did. Still, different approaches and definitions of functions are necessary to estimate the actual functional diversity of the microbiome.

## Figures and Tables

**Figure 1 microorganisms-09-00381-f001:**
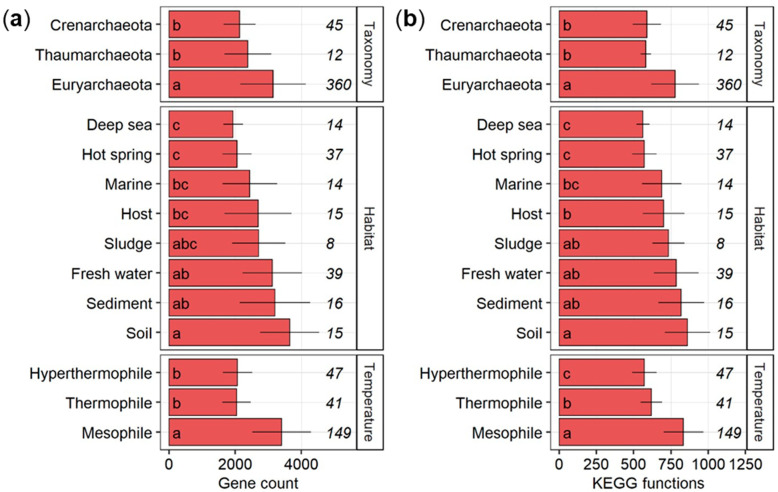
Counts (**a**) and the number of different KEGG functions (**b**) per genome across archaeal phyla, habitats, and temperature ranges shown as average with standard deviation. The number of archaeal genomes is given in italics. Groups followed by a different letter are significantly different according to the HSD-test (*p* < 0.05).

**Figure 2 microorganisms-09-00381-f002:**
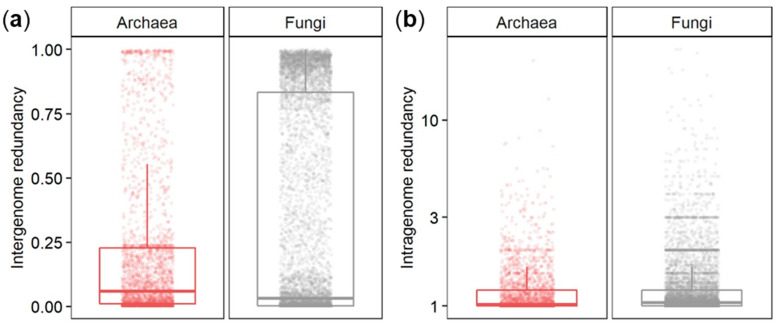
The distribution of intergenome functional redundancy as the total share of functions within archaea relative to the total number of archaeal species in the database (**a**) and intragenome functional redundancy as the number of replicated KEGG functions within one archaeal species in the database (**b**) compared to the previously published distributions in fungi [12].

**Figure 3 microorganisms-09-00381-f003:**
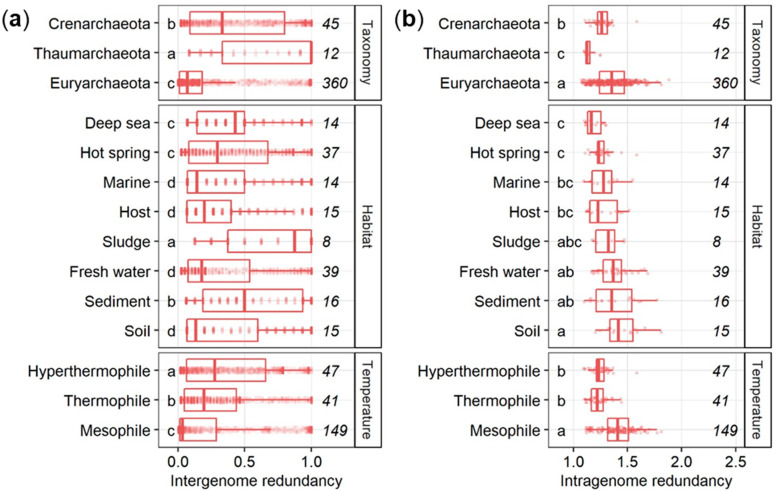
Intergenome (**a**) and intragenome functional redundancy (**b**) in archaeal phyla, habitats, and temperature ranges. The number of archaeal genomes is given in italic. Groups followed by a different letter are significantly different according to the HSD-test (*p* < 0.05).

**Figure 4 microorganisms-09-00381-f004:**
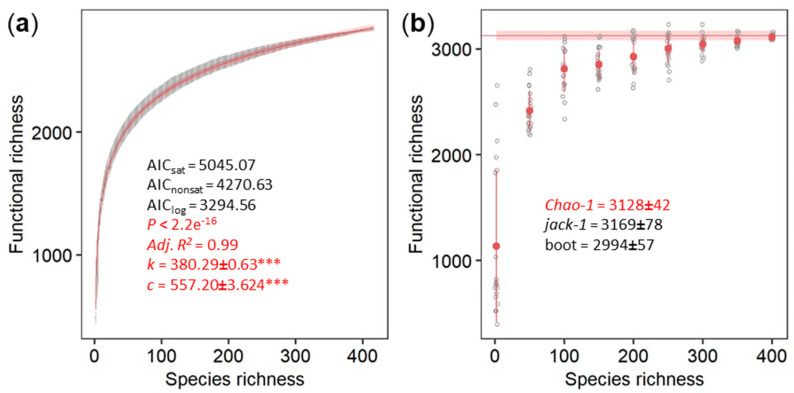
Parametric (**a**) and non-parametric (**b**) estimation of the total functional richness. The logarithmic model of the accumulation curves as gray points with error bars as 95% confidence intervals for the total known archaeal microbiome functions derived from the KEGG database by 1000 random permutations for every one species richness with 95% confidence intervals. Significance of the parameter estimates are indicated by asterisks (*** equals *p* < 0.001). The Chao-1 index as lower bound and non-parametric estimate was calculated using 20 replicates shown in gray for every 50 randomly picked archaeal genomes in the database starting with two species. The Chao-1 index of all 417 archaeal genomes with standard errors is shown as red line.

## Data Availability

Publicly available datasets were analyzed in this study. This data can be found here: https://img.jgi.doe.gov/.

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
