# Peer review of "Explorative Meta-Analysis of 417 Extant Archaeal Genomes to Predict Their Contribution to the Total Microbiome Functionality"

_microorganisms, 2021, doi:10.3390/microorganisms9020381_

Round 1

Reviewer 1 Report

The present article written by Starke et al. uses meta-analysis of multiple annotated extant and taxonomically unique archaeal genomes in order to predict their contribution to the total microbiome functionality. The study suggested a limited contribution of archaea to the functional potential of the total microbiome. While authors acknowledged the limitations of their study due to less genes covered in the KEGG dataset, their approach highlighted both similarities in functional redundancy as well as the difference in functional potential of archaea compared to other domains of life. This study will serve as a stepping stone for future advanced analysis of archaeal functional diversity estimation using datasets covering more genes in archaeal genomes. The article is scholarly and I recommend publication of the manuscript in the present form.

Author Response

Dear colleague,

we are very grateful for your positive feedback to our submission. We checked the English language with the aim to improve the grammar throughout the manuscript and highlighted changes in red in the revised version.

Thank you,
Robert Starke

Reviewer 2 Report

The authors have performed a meta-analysis on 417 available annotated genomes of archaea to fill in a gap relating the taxonomic and functional diversity of this domain. The paper is well written and contains interesting data. Three minor comments;

The importance of KEGG level 3 functions is demonstrated throughout the manuscript. For the readers who may not be as familiar as the authors are with KEGG, a simple sentence indicating just what a KEGG level 3 function is would be appreciated by the readers.

The authors use the term HSD-test (Honest Significant Difference) in the figure legends. Is this the Tukey HSD test?

Line 225, would 'habitats' be a better word choice here than 'niches'?

Author Response

Dear colleague,

we are very grateful for your positive feedback to our submission. We have checked the English language and aimed to improved the grammar throughout the manuscript with changes appearing in red in the revised manuscript. We further added a description for KEGG level 3 functions (L51-54), defined the used Tukey's HSD test in a separate paragraph for statistical analysis (L144-146), and replaced 'habitats' with niches in L229.